# Cardiotoxicity Induced by Protein Kinase Inhibitors in Patients with Cancer

**DOI:** 10.3390/ijms23052815

**Published:** 2022-03-04

**Authors:** Aleksandra Grela-Wojewoda, Renata Pacholczak-Madej, Agnieszka Adamczyk, Michał Korman, Mirosława Püsküllüoğlu

**Affiliations:** 1Department of Clinical Oncology, Maria Sklodowska-Curie National Research Institute of Oncology, Kraków Branch, Garncarska 11, 31-115 Kraków, Poland; renata.pacholczak@uj.edu.pl (R.P.-M.); mira.puskulluoglu@gmail.com (M.P.); 2Department of Anatomy, Jagiellonian University Medical College, 31-008 Kraków, Poland; 3Department of Tumour Pathology, Maria Sklodowska-Curie National Research Institute of Oncology, Kraków Branch, Garncarska 11, 31-115 Kraków, Poland; agnieszka.adamczyk@onkologia.krakow.pl; 4Faculty of Medicine, Jagiellonian University Medical College, 31-008 Kraków, Poland; michal.korman@student.uj.edu.pl

**Keywords:** cardiotoxicity, molecular mechanisms of cardiac damage, protein kinase inhibitors, targeted treatment, cancer

## Abstract

Kinase inhibitors (KIs) represent a growing class of drugs directed at various protein kinases and used in the treatment of both solid tumors and hematologic malignancies. It is a heterogeneous group of compounds that are widely applied not only in different types of tumors but also in tumors that are positive for a specific predictive factor. This review summarizes common cardiotoxic effects of KIs, including hypertension, arrhythmias with bradycardia and QTc prolongation, and cardiomyopathy that can lead to heart failure, as well as less common effects such as fluid retention, ischemic heart disease, and elevated risk of thromboembolic events. The guidelines for cardiac monitoring and management of the most common cardiotoxic effects of protein KIs are discussed. Potential signaling pathways affected by KIs and likely contributing to cardiac damage are also described. Finally, the need for further research into the molecular mechanisms underlying the cardiovascular toxicity of these drugs is indicated.

## 1. Introduction

The incidence of cancer has been increasing worldwide. Data show that cancer may become the leading cause of death in some countries in the 21st century. In 2017, there were 24.5 million new cancer cases and 9.6 million cancer-related deaths, as compared with 12.7 million and 7.4 million, respectively, in 2008 [1,2]. In women, the most common type of cancer is breast cancer, followed by lung, cervical, and colorectal cancer [2]. In men, the highest incidence was reported for lung cancer, followed by colorectal, gastric, and prostate cancer [2]. The incidence of cancer depends on the country, the advancement of climate change, socioeconomic conditions, the level of education, lifestyle and life expectancy, and the organization of the healthcare system [3,4]. 

Recently, there has been significant progress in oncologic treatment. For many years, chemotherapy and hormone therapy have been the major options for systemic cancer treatment. However, advances in the field of basic sciences and improved understanding of carcinogenesis have led to the development of modern agents that are now widely available, namely, molecularly targeted drugs and immunotherapy. Thus, the current options for systemic cancer treatment include chemotherapy, hormone therapy, targeted drugs, and immunotherapy, commonly used in combination. Among targeted therapies, protein kinase inhibitors (KIs) are a growing group of drugs used in the majority of cancer indications, including lung, breast, colorectal, kidney, liver, or thyroid cancer, but also melanoma, soft tissue sarcoma, gastrointestinal stroma tumor (GIST), and various hematological malignancies [5,6] New strategies have changed the treatment paradigm in many types of cancer, which are now considered to be chronic diseases rather than fatal diseases with a short life expectancy. However, new treatment modalities, including protein KIs come at the cost of unavoidable side effects, including cardiotoxicity. 

Chemotherapy and molecularly targeted drugs can cause cardiac dysfunction, which affects patient survival and the quality of life. The most common type of cardiotoxicity is symptomatic or asymptomatic left ventricular (LV) dysfunction, which can be induced both by classic chemotherapy (such as anthracyclines) and by molecularly targeted drugs [7,8,9]. The monitoring of left ventricular function in clinical practice is based on the evaluation of left ventricular ejection fraction (LVEF) during echocardiographic examination and electrocardiogram (ECG) recording.

Cardiovascular diseases and cancer are the most common health problems in highly developed countries. The optimal strategy for modern oncologic treatment should be based on a comprehensive approach to oncologic and cardiology patients. 

The aim of this review was to explore molecular mechanisms of cardiotoxicity induced by protein KIs in patients with cancer and to summarize the frequency and clinical significance of the most common cardiac toxicities caused by KIs.

## 2. Families of Protein Kinases

Protein kinases are enzymes responsible for the phosphorylation of proteins that are involved in various signaling pathways. Phosphorylation results in cell proliferation, migration, survival, and differentiation. Thus, any disturbances in this process are key elements of carcinogenesis and result in the transformation of the proto-oncogene into the oncogene. The discovery of KIs has changed the paradigm of treatment for various malignancies. As of January 2022, the Food and Drug Administration and the European Medicines Agency have approved more than 60 KIs for use in the therapeutic setting.

Protein KIs can be classified into several families depending on the structure and ligand: epidermal growth factor receptor (EGFR) inhibitors; vascular endothelial growth factor receptor (VEGFR) tyrosine kinase inhibitors; v-raf murine sarcoma viral oncogene homolog B (BRAF) 1/2 inhibitors; mitogen-activated protein kinase kinases 1 and 2 (MEK) inhibitors; anaplastic lymphoma kinase (ALK) inhibitors; Bruton tyrosine kinase (BTK) inhibitors; phosphoinositide 3-kinase (PI3K) inhibitors; cyclin-dependent kinase (CDK) 4/6 inhibitors; Janus kinases (JAK) inhibitors; BCR-ABL tyrosine kinase inhibitors; FMS-like tyrosine kinase-3 (FLT3) inhibitors; and stem cell factor receptor/platelet-derived growth factor receptor (KIT/PDGFRA) inhibitors (Figure 1 and Figure 2). Specified molecular targets and therapeutic indications currently available in the European Union are presented in Table 1 [5,10]. 

Additionally, KIs are classified according to their biochemical properties and specific mechanism of action. This classification has evolved over the decades. Initially, Dar and Shokat [6] proposed three types of inhibitors depending on the way they bind to the adenosine triphosphate pocket (i.e., to its active conformation type I, inactive type II, and non–adenosine triphosphate competitive inhibitor or allosteric inhibitor type III). Subsequently, allosteric inhibitors were divided into two classes (type III (allosteric inhibitors) and type IV (substrate-directed inhibitors)) [11]. Lambda and Gosh [12] distinguished also type V inhibitors, which span two regions of the protein kinase domain, and type VI inhibitors, which form covalent adducts. In this review, we classified KIs according to the Roskoski classification, but the subcategories were omitted for simplicity [13]. Some of the KIs were thus unclassified and were labeled as “no data” in Table 1. 

## 3. Cardiotoxicity of Oncologic Drugs

Cardiac dysfunction related to cancer therapies has several manifestations, from an asymptomatic reduction in LVEF to chronic heart failure. Classic chemotherapy leads to irreversible cardiomyocyte damage [7,25]. The mechanism of anthracycline-induced irreversible heart damage has been extensively studied. The analysis of myocardial biopsy enhanced the understanding of the structural damage caused by these drugs. The formation of free radicals and the induction of oxidative stress destroy the myocardial ultrastructure—microfibril architectonics is disrupted, and the vacuolation and apoptosis of cardiac cells are observed. Initially, these changes may be asymptomatic; however, in some patients, symptoms of heart failure can develop dynamically even many years after the end of the oncologic treatment. Nevertheless, modern molecularly targeted agents (e.g., antiangiogenic KIs) act by changing the energetic pathways of cardiomyocytes and the metabolism of contractile proteins, usually resulting only in temporary heart dysfunction. The clinical consequences of this transitory disorder remain unclear [17,26]. Four types of cardiotoxicity have been described [8,9,27]: 

Acute cardiotoxicity—It occurs rarely, manifests immediately after the first drug administration, and is dose-independent. Its symptoms include hypotony, arrhythmias, and myocardial ischemia. It is usually reversible after discontinuation of drug infusion.

Subchronic cardiotoxicity—It occurs rarely, and its onset is observed in the first weeks of treatment with high doses of drugs. It usually manifests with myocarditis or pericarditis (e.g., after anthracyclines administration).

Early-onset chronic cardiotoxicity—It develops within a few weeks after discontinuation of treatment and manifests as progressive heart failure.

Late-onset chronic cardiotoxicity—It develops many years after the end of the treatment and leads to heart failure.

A new classification of chemotherapy-related cardiac dysfunction (CRCD) was established after the introduction of modern oncologic agents [28]:

Type I CRCD—irreversible (e.g., after anthracycline administration)

Type II CRCD—potentially reversible, induced by new-generation human epidermal growth factor receptor 2 (HER2)-targeted agents and KIs and is potentially reversible.

Chemotherapy usually comprises several cytostatic agents which might be administered concurrently with targeted agents; therefore, oncologists should consider their possible synergistic toxicity.

## 4. Molecular Mechanisms of Cardiotoxicity of Selected Kinase Inhibitors

The mechanism of KI-induced cardiotoxicity is not fully understood. Examples of KIs with their cardiotoxic effects and the underlying molecular and cellular pathways are presented in Table 2 [29,30,31,32,33]. Generally, the mechanisms of cardiac toxicity can be divided into on-target and off-target mechanisms [33]. On-target mechanisms occur when the inhibition of a molecule by KIs causes an anticancer effect in malignant cells but also leads to toxicity in normal cells. On the other hand, off-target mechanisms occur when KIs inhibit one kinase in malignant cells, causing an anticancer effect, and other kinases in normal cells, leading to cardiac toxicity.

KIs (especially tyrosine kinase inhibitors [TKIs]) inhibit numerous molecular pathways, and various factors contribute to the induction of hypertension. One of the hypotheses is that the inhibition of vascular endothelial growth factor and platelet-derived growth factor receptor (PDGFR) activity results in reduced production of nitric oxide (NO) in the vascular endothelium. On the other hand, a decrease in endothelial NO concentrations leads to a disruption of the endothelin function and an increase in peripheral resistance, which induces a rise in blood pressure [44,45,46]. A decrease in urinary nitrite excretion and a decrease in the concentration of NO metabolites were observed in patients treated with TKIs, which supports this theory [47,48]. There are ongoing studies evaluating the role of other vasoactive factors in inducing arterial hypertension in patients treated with KIs. These other factors include prostaglandins, the renin-angiotensin-aldosterone system, thromboxane, and the effect of KIs on sympathetic nervous system stimulation [47,48,49]. 

KIs affect the cardiac conduction system by blocking the cardiac channels, leading to bradyarrhythmia (e.g., atrioventricular blocks, QTc prolongation) and tachyarrhythmia (e.g., atrial fibrillation, supraventricular tachycardia) [32,36,50]. One of the on-target mechanisms underlying the development of arrhythmia can be PI3K blockage. It leads to a rise of persistent sodium current and a reduction in L-type calcium and potassium current as well as QTc prolongation [33]. An example of an off-target mechanism leading to arrhythmia is ibrutinib-induced inhibition of Tec protein tyrosine kinase responsible for cardioprotective function by regulating the PI3K-Akt pathway [29,33].

Left ventricular dysfunction and heart failure may occur as a result of mitochondrial damage, alterations in cardiac energy balance, and contractile protein dysfunction [7,25]. According to one hypothesis, the inhibition of PDGFR and other tyrosine kinase receptors in cardiomyocytes, which determines their functioning and survival, disrupts the normal response of the myocyte to hypertensive stress [51,52]. Other authors emphasized the importance of the ribosomal S6 kinase family, as it determines cardiomyocyte survival by inhibiting the phosphorylation of apoptosis-activating factors (such as the proapoptotic protein BAD or the activated protein kinase AMPK). By interfering with this molecular pathway, KIs may promote cardiac damage [53]. The inhibition of the KIT and RAF1 pathways leads to vascular stem cell damage and endothelial dysfunction [54]. Endothelial cell apoptosis and exposure of subendothelial collagen initiate the coagulation process. Thus, on the one hand, endothelial dysfunction activates the coagulation process, leading to thromboembolic episodes, and, on the other hand, it destabilizes the atherosclerotic plaque, leading to ischemic episodes. In addition, a genetic predisposition in the form of *RAF1* deletion leads to increased myocyte apoptosis, worse contractility, and enhanced left ventricular fibrosis and dilatation. The inhibition of the kinase pathway by KIs is associated with lower myocardial vascular density, which promotes cardiac fibrosis and reduces myocardial contractility [55]. These disorders are also initiated by induced arterial hypertension [56]. 

The mechanism of fluid retention caused by imatinib remains unclear. The blockage of PDGFR signaling that is responsible for the homeostasis of interstitial fluid was suggested as one of the mechanisms [57].

The molecular mechanisms underlying KI-induced cardiotoxicity are complex and not fully understood (e.g., an unclear mechanism of hypertension caused by copanlisib [38] or qTc prolongation due to inhibition of CDK4/6 [43]). On the one hand, various KIs can induce comparable cardiotoxic effects via similar mechanisms (e.g., promoting hypertension through the inhibition of NO formation by sorafenib or ibrutinib [29,34]). On the other hand, one KI can promote specific cardiotoxicity by way of a few molecular mechanisms (e.g., activation of numerous mechanisms leading to hypertension induced by TKIs [34]).

## 5. Cardiotoxicity of Protein Kinase Inhibitors

Protein KIs can lead to numerous cardiovascular toxicities including, but not limited to, hypertension, arrhythmias, cardiomyopathy or heart failure, fluid retention, thromboembolic events, and myocardial ischemia or infarction [40,41,42,43,44,45,46,53,54,57,58]. Considering the toxicity profile of KIs, the National Cancer Institute recommends that before starting therapy, each patient should be assessed for cardiovascular risk factors such as previous cardiovascular disease, blood pressure above 160/100 mmHg, diabetes, dyslipidemia, smoking, obesity (body mass index >30 kg/m^2^), lack of physical activity, alcohol abuse, excessive salt consumption, and family history of cardiovascular disease [7,58]. Standard assessment of cardiac function is performed with ECG. In some patients, echocardiography with LVEF assessment and intima–media complex measurement by carotid ultrasound are used [58,59,60].

### 5.1. Hypertension

Sunitinib, sorafenib, lenvatinib, axitinib, pazopanib, and cabozantinib are KIs used in the treatment of renal cell carcinoma (RCC) [34,61]. In clinical trials, all these drugs were associated with hypertension [34,62]. A trial investigating the use of adjuvant sunitinib in patients with RCC showed hypertension in 44.7% of patients vs. 13.1% in the placebo group [63]. In a phase III clinical trial on metastatic RCC (mRCC), 17% of the 903 participants receiving sorafenib developed hypertension, as compared to 1% in the placebo group [64]. A comparison of lenvatinib alone, everolimus alone, and a combination of the two agents in a phase II clinical trial on mRCC revealed that lenvantinib induces hypertension both when used alone and when used in combination (48% vs. 41% vs. 10%, respectively). In a phase II trial comparing axitinib and sorafenib, axitinib showed a higher toxicity profile, with 40% of patients showing any grade of hypertension (vs. 29% in the sorafenib group). However, this adverse event led to treatment discontinuation in only one patient in the axitinib group [45].

In a phase II clinical trial including patients with non-small cell lung cancer (NSCLC) and ALK rearrangement, brigatinib resulted in elevated blood pressure in 20% of the participants [65]. 

Ibrutinib is a BTK inhibitor used in hematologic B-cell malignancies. A large observational study of ibrutinib-treated patients estimated the risk of hypertension at around 2%. The incidence of hypertension was higher in a clinical trial including 562 patients. It was reported in almost 80% of participants, of whom 18% developed grade hypertension according to the Common Terminology Criteria for Adverse Events (CTCAE), defined as a blood pressure >160/100 mmHg [66,67]. There are scarce data on hypertension in patients treated with zanubrutinib, a drug approved for use in patients with refractory mantle cell lymphoma. It seems that zanubrutinib has a lower risk of elevated blood pressure than ibrutinib, with the rate of grade 3 hypertension reported at around 5% in a pooled safety analysis of six studies and of almost 12% (when evaluated as adverse events of special interest) in another pooled analysis of two studies on refractory mantle cell lymphoma [68]. 

Another KI, selpercatinib, caused grade 3 or 4 hypertension in 21% of patients enrolled in phase I-II trial of thyroid cancer [69]. Similarly, in a population with NSCLC positive for rearranged during transfection (RET)-fusion, selpercatinib induced grade 3 or 4 hypertension in 14% of patients [70]. 

Other KI families associated with increased blood pressure include PI3K inhibitors and JAK inhibitors. An interesting example of PI3K inhibitors is copanlisib, a pan-class I PI3K inhibitor that caused hypertension of any grade in 54.8% of patients enrolled in a phase II clinical trial, and grade 3 in 40.5% of patients, usually within the first 2 h after the first infusion [32,71]. In contrast, one of the JAK inhibitors, ruxolitinib, resulted in increased systolic blood pressure after approximately 72 weeks of treatment [32,72]. 

Elevated blood pressure is observed most often in the first weeks of treatment [73]. Management strategies for TKI-induced hypertension according to CTCAE grades are presented in Table 3 [7,58].

Hypertension that occurs in patients treated with TKIs (e.g., sunitinib or sorafenib) is a predictor of treatment responses. Studies indicate that patients who develop hypertension during TKI therapy have a longer median overall survival and progression-free survival compared with those who do not develop this type of cardiotoxicity [46,74,75]. The latter group requires adequate and effective cardiovascular therapy.

### 5.2. Arrhythmias

#### 5.2.1. Bradycardia

Oral inhibitors of ALK-positive tumors (e.g., crizotinib and ceritinib) were reported to cause sinus bradycardia. Trials on crizotinib treatment of NSCLC revealed a bradycardia frequency between 0.5% and 70%, with the average time to the lowest heart rate after 18.6 weeks [76,77]. However, the reported incidence of this adverse event depends on the definition of bradycardia. In most cases, it is defined as asymptomatic and not related to other arrhythmias [76]. A meta-analysis suggested that the new generation of ALK inhibitors such as alectinib, brigatinib, or lorlatinib was associated with a similar risk of bradycardia when compared with crizotinib (estimated at around 8%), while the risk was lower during the ceritinib treatment [78]. Therefore, it is suggested that these drugs should be avoided in patients with preexisting bradycardia and patients who use beta-blockers or other antiarrhythmic medications. The guidelines recommend regular heart rate monitoring with appropriate dose adjustments [79].

#### 5.2.2. QTc Prolongation

Sunitinib and other KIs used in the treatment of mRCC have been reported to prolong QTc, an effect that seems to be dose-dependent. KIs used in this setting can also cause other conduction disturbances that lead to ECG abnormalities [53,80,81]. A meta-analysis that included 6548 patients from 18 clinical trials showed a significant risk of QTc prolongation for sunitinib but not for pazopanib or axitinib and sorafenib [80,82].

Selpercatinib (used in RET-fusion positive medullary thyroid cancer and NSCLC) was also shown to prolong QTc, with the reported rates of 6% for prolongation >500 ms and 15% for prolongation ≥60 ms from baseline. In a phase I-II clinical trial on NSCLC, selpercatinib prolonged QTc in 10% of participants [70]. Another RET inhibitor that prolongs QTc and causes torsade de pointes is vandetanib. A meta-analysis of phase II and III clinical trials revealed QTc prolongation (all grades) in 16.4% of patients. In patients with thyroid cancer, who required longer treatment, the incidence increased to 18% [83,84,85]. Another agent in this class is lenvatinib with QTc prolongation (any grade) reported in 9% of participants (vs. 2% in the placebo group) and grade ≥3 in 2% of participants (vs. 0% in the placebo group). 

In the case of KIs used in the treatment of ALK-positive NSCLC, crizotinib and ceritinib, the risk of QTc prolongation does not appear to exceed 1–3% of the treated population [85,86]. 

It was suggested that EGFR inhibitors, including osimertinib and mobocertinib, also affect QTc, especially in patients with underlying conditions predisposing to QTc prolongation. A pooled safety analysis including 250 patients showed QTc prolongation of >500 ms from baseline in 1.2% of patients and >60 ms from baseline in 11%. Cases of torsades de pointes were also associated with this treatment [87]. 

Additionally, BRAF inhibitors, such as vemurafenib and encorafenib, were shown to prolong the QTc interval. In a recent meta-analysis that included five trials of patients with metastatic melanoma, the proportion of patients experiencing QTc prolongation was not higher than when a MEK inhibitor was added to monotherapy with BRAF inhibitors [40].

Another class of drugs includes CDK4/6 inhibitors such as ribociclib, abemaciclib, and palbociclib. In a large meta-analysis of 6 clinical trials with a total of 3743 patients, palbociclib showed a significantly lower risk of QTc prolongation than ribociclib [88]. For ribociclib, the prolongation occurred during the first treatment cycle [89]. The combination of ribociclib with letrozole seems to be associated with a lower incidence of QTc prolongation than its combination with tamoxifen, as shown in the MONALEESA-7 trial (16% vs. 7%, respectively) [90].

Finally, among the KIs used for the treatment of Philadelphia chromosome-positive chronic myelogenous leukemia, bosutinib, dasatinib, and nilotinib are associated with QTc prolongation [91,92]. 

Considering these cardiotoxic effects, the above KIs should be administered with caution. Regular ECG monitoring with QTc interval measurement is recommended at baseline and during treatment. Additionally, thyroid hormones and electrolytes, including potassium, magnesium, and calcium, should be monitored regularly and corrected in the case of abnormalities. There are several formulas used for QTc interval calculations, including the Bazett, Fridericia, and Framingham calculators [91,93,94,95]. None of them have shown to be superior over the other. Any concomitant medications that could influence the QTc interval should be administered with caution. These include CYP3A4 inhibitors that increase KIs activity. Patients with preexisting conditions, including a history of QTc prolongation (>450 ms), bradycardia, and hyperthyroidism, and with electrolyte disorders are also at higher risk for QTc prolongation. Any electrolyte abnormalities should be corrected, and the treatment should be interrupted if the QTc interval is greater than 500 ms [91,92,94,95,96].

#### 5.2.3. Other Arrhythmias

Ibrutinib is a well-known agent that causes supraventricular and ventricular arrhythmias (7% and 0.5% of patients, respectively, as revealed by a safety report for 13,572 patients). In another series of more than 500 patients, the incidence of atrial fibrillation was 13% [67]. The risk of atrial flutter and atrial fibrillation associated with zanubrutinib treatment appears to be lower (2% in clinical trials) [68]. Conduction disturbances, rhythm abnormalities, and variations in the QRS amplitude or QRS axis are quite common ECG findings in patients receiving various KIs [33,91,92,94,95,96].

### 5.3. Cardiomyopathy and Heart Failure

Cardiomyopathy leading to heart failure is a serious adverse effect of treatment with various classes of KIs, including BCR-ABL kinase (e.g., dasatinib, imatinib), c-KIT (ripretinib, imatinib), BRAF/MEK, EGFR, and ALK inhibitors, as well as multitarget KIs used for mRCC treatment. It was reported that 1.6% of patients receiving dasatinib develop serious cardiac toxicity, including cardiomyopathy and heart failure [97,98].

For imatinib, the risk was reported to be 1–2% for clinically significant heart failure in patients with hematologic malignancies. On the other hand, the risk was not significant in patients with GIST, although elevated serum levels of brain natriuretic peptide in those patients suggest subclinical changes [35,97,98,99,100,101,102]. In a phase III clinical trial, ripretinib caused LVEF reduction in 2.6% of patients and heart failure in 1.2%. In a pooled safety analysis of over 350 patients, the risk of heart failure was 1.7% (for any grade, including also cases of diastolic dysfunction and ventricular hypertrophy) [103,104]. 

BRAF inhibitors are associated with a higher risk of cardiomyopathy when combined with MEK inhibitors. During trametinib treatment for metastatic melanoma, 11% of patients were shown to develop left ventricular dysfunction. In a phase III trial, reduced LVEF occurred in 5.2% of patients. In these patients, cardiomyopathy resolved in most cases, allowing treatment continuation [105,106]. 

Osimertinib, an EGFR TKI, was reported to reduce LVEF in about 3–5% of patients in clinical trials and a pooled analysis. Most of these patients were found to be asymptomatic and did not require discontinuation of treatment. For mobocertinib, the risk seems to be similar, with a frequency of heart failure in a pooled safety analysis of 250 patients estimated at 2.7% [107,108,109]. 

Sorafenib, sunitinib, pazopanib, axitinib, and lenvatinib are multitarget KIs used in mRCC and other malignancies. These drugs are also associated with the risk of heart failure. In a large meta-analysis of numerous KIs directed against VEGFR, heart failure was reported in 2.4% of patients vs. 0.8% of the placebo group [82]. Another meta-analysis of almost 7000 patients receiving sunitinib showed a 4.1% incidence of heart failure of any grade [110]. However, data from clinical trials and retrospective studies revealed a decrease in LVEF in 21–28% of patients, with a heart failure incidence of 3–15% [110,111,112]. In patients with soft tissue sarcomas, pazopanib showed a significant decrease in LVEF compared with placebo (6.7% vs. 2.4%, respectively) [113]. Similar rates were observed for lenvatinib in patients with advanced thyroid cancer [114,115]. 

The HER2 inhibitor lapatinib is used in metastatic breast cancer. Although its cardiac toxicity has not been studied as extensively as that of the anti-HER2 monoclonal antibody trastuzumab, a meta-analysis including 26 studies revealed left ventricular dysfunction in 1.6% of patients and LVEF reduction in 2.2% [116]. 

For the majority of the above inhibitors, regular echocardiographic evaluation of LVEF at baseline and during treatment is recommended. The risk of LVEF decline and heart failure is believed to be underestimated in clinical trials. However, in the case of inhibitors with lower or unclear risk (e.g., imatinib), monitoring is required only in patients with predisposing conditions or symptoms suggesting cardiac disease. Initiation of treatment in patients with LVEF <50% should be avoided, and discontinuation of treatment is necessary in case of LVEF reduction [10,117,118,119].

### 5.4. Fluid Retention

Some KIs can cause fluid retention, leading to pulmonary edema or pericardial effusion. This symptom is observed during imatinib or bosutinib treatment, but it rarely leads to heart failure, according to existing data (about 1% of patients). Practical guidelines published in 2018 recommend standard treatment with diuretics and supportive care with dose reduction or discontinuation of treatment if needed [32,120,121]. Fluid retention with pulmonary edema was described also for dasatinib use [32,122]. Other drug classes associated with fluid retention are BCR-ABL inhibitors, while the FLT3 inhibitor gilteritinib was reported to cause edema of any grade in 24% of patients [32,123].

Patients with fluid retention should be assessed for cardiac, liver, kidney, and thyroid function, as well as serum protein and albumin levels. Moreover, ultrasound is recommended to diagnose and monitor patients suspected of pulmonary edema, pericardial effusion, ascites, and pleural effusion [57,122].

### 5.5. Arterial Thromboembolic Events

A large systematic review with a meta-analysis of 10,000 patients included in 10 phase II and III studies on sunitinib and sorafenib revealed a prevalence of arterial thromboembolic events of 1.4%, with no difference between the drugs [124]. These events can also be attributed to other multitarget KIs used in mRCC, as revealed by clinical trials for pazopanib (3% of patients vs. 0% in the placebo group) and for lenvatinib (5% of patients vs. 2% in the placebo group) [125,126,127].

Existing guidelines underline the importance of prevention and prompt identification and treatment of patients. Predisposing factors should be managed before the initiation of therapy. In the case of a thromboembolic event, treatment should be discontinued, and the condition should be treated according to applicable guidelines [128,129,130]. Any decision on treatment continuation must be considered on an individual basis. 

### 5.6. Venous Thromboembolic Events

None of the two recent meta-analyses confirmed an elevated risk of venous thromboembolic events in patients treated with sunitinib, sorafenib, axitinib, pazopanib, and vandetanib [126,131]. The administration of anticoagulant prophylaxis during the antiangiogenic KI therapy in ambulatory patients in the absence of other risk factors for venous thromboembolic events is not recommended. 

### 5.7. Myocardial Ischemia and Infarction

Some KIs are suspected to increase the risk of cardiac ischemia. For regorafenib, the incidence of cardiac ischemia in patients with metastatic colorectal cancer was 1.2% (vs. 0.4% in the placebo group) [132]. Another study reported an association with type 5 myocardial infarction [132,133]. For sorafenib, the incidence of cardiac ischemia was estimated at 2.9% (vs. 0.4 in the placebo group) and 2.7% (vs. 1.3% in the placebo group) [91,134]. Finally, ibrutinib use was associated with an incidence of cardiac ischemia of 1.4% in a study with over 500 patients with B-cell malignancies [67]. The importance of monitoring cardiac enzymes and ECG abnormalities remains uncertain [102]. 

Patients with coronary artery disease should be identified before initiation of therapy. The algorithm used to diagnose coronary artery disease in patients with cancer treated with KIs does not differ from that used in patients without cancer [135].

## 6. Conclusions

According to the literature, KIs therapy is associated with the risk of cardiotoxicity. Clinical trials reported that cardiotoxicity manifests itself as hypertension, heart failure, cardiac arrhythmias (bradyarrhythmias, tachyarrhythmias, qTc prolongation), thromboembolic events, fluid retention, and exacerbation of coronary artery disease. 

Importantly, the assessment of cardiac dysfunction and cardiac adverse events was not the primary or secondary endpoint in any of the later-phase clinical trials described in Section 5; these outcomes were assessed retrospectively. The discrepancies in the reported frequency of cardiovascular complications may be due to differences in the definition of the same phenomenon across individual studies or the actual differences in the frequency and severity of these cardiac complications. Only studies specifically aimed at assessing cardiac dysfunction could provide comprehensive and sufficient data to develop clinical guidelines for the prevention and management of KI-induced cardiotoxicity.

The predictors and mechanisms of cardiotoxicity are yet to be defined in future research. The current gaps in knowledge result from the relative novelty of this specific drug class. The number of preclinical studies exploring the molecular mechanisms of KI activity and toxicity has been growing rapidly, providing more data but also some more questions. KIs are often multitargeted drugs, with new targets being commonly identified after the drug has been introduced to the market [136]. This multifunctionality poses a challenge in terms of developing a comprehensive classification system for these drugs [13].

Clearly, cardiotoxicity induced by KIs is affected by classic cardiovascular risk factors. Therefore, these risk factors should always be evaluated before initiation of therapy and managed accordingly. Cancer patients treated with KIs should undergo regular cardiac monitoring, including echocardiography, electrocardiography, and, in some patients, biochemical tests. Understanding the mechanisms and predictors of cardiotoxicity resulting from systemic anticancer treatment may allow adequate prevention of cardiovascular complications as well as a reduction in their incidence [137].

## Figures and Tables

**Figure 1 ijms-23-02815-f001:**
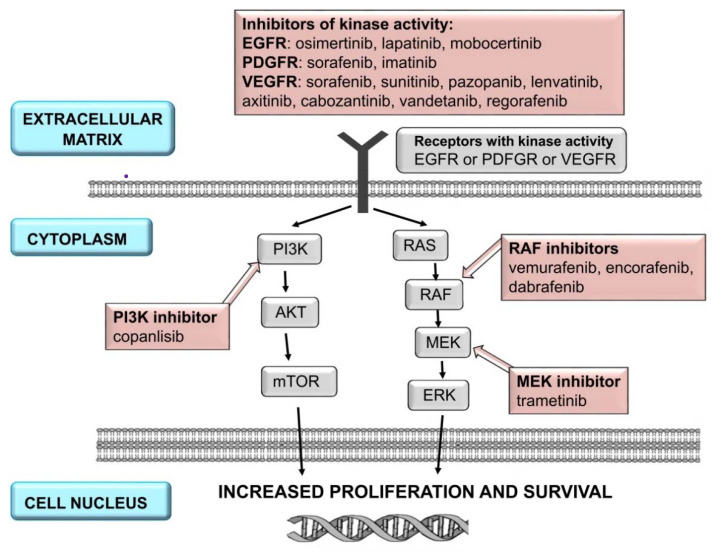
A simplified scheme of signal transmission in cell and selected target points: EGFR, MEK, PGDFR, PI3K, RAF, and VEGFR for molecularly targeted drugs (protein kinase inhibitors) [14,15,16,17,18].

**Figure 2 ijms-23-02815-f002:**
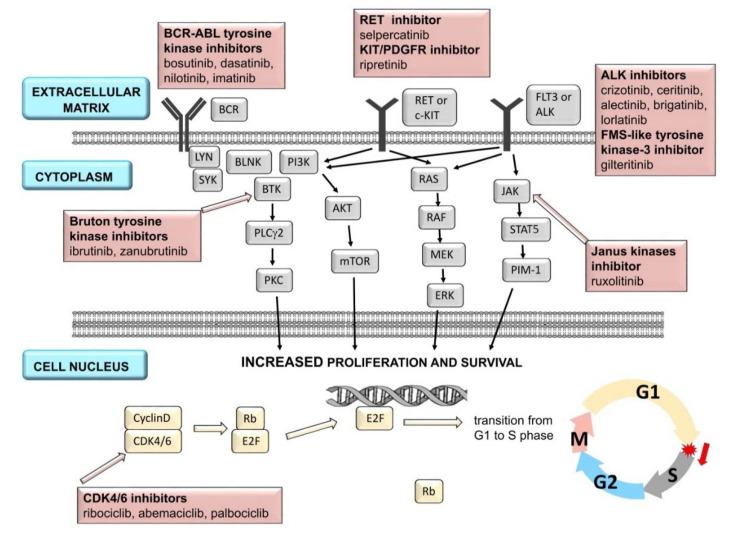
A simplified scheme of signal transmission in cell and selected target points: ALK, BCR-ABL, Bruton tyrosine kinase, CDK 4/6, FMS-like tyrosine kinase-3, Janus kinases, KIT/PDGFR, and RET for molecularly targeted drugs (protein kinase inhibitors) [19,20,21,22,23,24].

**Table 1 ijms-23-02815-t001:** Molecular targets and therapeutic indications for protein kinase inhibitors approved for use in adult patients with cancer by the European Medicines Agency (EMA) unless otherwise specified (compiled on the basis of the Summary of Product Characteristics as of January 2022).

Inhibitor	Subtype	Molecular Targets	Therapeutic Indications	Type and Incidence of Cardiotoxicity ^&^	Class ^
Vascular endothelial growth factor receptor tyrosine kinase inhibitors					
**Sunitinib**	Receptor TKI	PDGFRα, PDGFRβ, VEGFR-1, -2, -3, KIT, FLT3, CSF-1R, RET	GIST after treatment failure/imatinib intolerance mRCC—metastatic setting pNET—unresectable/metastatic	Hypertension (44.7%) QTc prolongation (0.5%) # Cardiomyopathy (3–15%) Arterial and venous thromboembolic events (1.4%)	I/II
**Pazopanib**	Receptor TKI	PDGFRα, PDGFRβ, VEGFR-1, -2, -3, KIT	mRCC—first-line treatment or after prior cytokine therapy Advanced soft tissue sarcomas—second-line treatment or progression within 12 months after prior (neo)adjuvant therapy (certain subtypes)	Hypertension (41%) ^#^ QTc prolongation (<1%) Cardiomyopathy (6.7%) Arterial and venous thromboembolic events (3%)	I
**Lenvatinib**	Receptor TKI	PDGFRα, VEGFR-1, -2, -3, KIT, RET, FGFR-1, -2, -3, -4	Differentiated thyroid carcinoma refractory to radioactive iodine Advanced HCC—first-line treatment Advanced endometrial carcinoma—with pembrolizumab, after prior platinum-containing chemotherapy	Hypertension (48%) QTc prolongation (9%) Cardiomyopathy (<1%) ^#^ Arterial and venous thromboembolic events (5%)	I/II
**Sorafenib**	Multikinase inhibitor	PDGFRß, VEGFR-2, -3, CRAF, BRAF, KIT, FLT3	HCC—metastatic setting mRCC—after prior interferon alfa/interleukin-2 therapy or unsuitable for such treatment Differentiated thyroid carcinoma refractory to radioactive iodine	Hypertension (17–29%) QTc prolongation (rare) ^#^ Cardiomyopathy (rare) ^#^ Arterial and venous thromboembolic events (1.4%) Myocardial ischemia (2.9%)	II
**Axitinib**	Receptor TKI	VEGFR-1, -2, -3	mRCC—after prior sunitinib or cytokine therapy	Hypertension (40%) QTc prolongation (rare) ^#^ Cardiomyopathy (1.8%) ^#^ Arterial and venous thromboembolic events (2.8%) ^#^	II
**Cabozantinib**	Receptor TKI	MET, VEGFR, AXL, RET, ROS1, TYRO3, MER, KIT, TRKB, FLT3, TIE-2	mRCC: - first-line treatment as a single agent in intermediate/poor-risk patients - first-line treatment in combination with nivolumab - second-line treatment after prior anti-VEGF therapy HCC—after prior sorafenib treatment	Hypertension (>25%) ^#^ Arterial and venous thromboembolic events (1–10%) ^#^	I
**Vandetanib**	Receptor TKI	VEGFR-2, EGFR, RET	Advanced medullary thyroid cancer	Hypertension (>10%) ^#^ QTc prolongation (16.4–18%) Cardiomyopathy (>0.1%) ^#^ Arterial and venous thromboembolic events (>1%) ^#^	I
**Regorafenib**	Multikinase inhibitor	VEGFR-1, -2, -3, TIE-2, PDGFR, FGFR, KIT, RET, RAF-1, BRAF, CSF1R	mCRC—progression after available therapies GIST—after prior imatinib and sunitinib HCC—after prior sorafenib	1. Myocardial ischemia (1.2%)	II
**Bruton tyrosine kinase inhibitors**					
**Ibrutinib**	Bruton TKI	BTK	MCL—relapsed/refractory CLL: - first-line treatment as a single agent or with rituximab/obinutuzumab - subsequent lines as a single agent or in combination with rituximab and bendamustine Waldenström macroglobulinemia—first-line treatment for patients unsuitable for chemo-immunotherapy or subsequent lines	Hypertension (2–80%) Atrial fibrillation (13%) Ventricular arrhythmias (1%) ^#^ Myocardial ischemia (1.4%)	VI
**Zanubrutinib ***	Bruton TKI	BTK	MCL—after at least two prior therapies	Hypertension (5–12%) Atrial flutter/fibrillation (2%)	n/d
**Phosphoinositide 3-kinase inhibitors**					
**Copanlisib ***	PI-3K kinase inhibitor	PI-3K	Follicular lymphoma—after at least one prior therapy	Hypertension (54.8%)	n/d
**Janus kinases inhibitors**					
**Ruxolitinib**	Janus kinases inhibitor (JAKs)	JAK-1, -2	Myelofibrosis Polycythemia vera—after prior hydroxyurea treatment	Hypertension (>10%) ^#^	I
**Anaplastic lymphoma kinase inhibitors**					
**Crizotinib**	ALK—receptor TKI	ALK, HGFR, ROS-1	NSCLC—ALK-positive in the first or subsequent lines of treatment NSCLC—ROS-1 positive	Bradycardia (0.5–70%) QTc prolongation (1–3%) Cardiomyopathy (1%) ^#^	I
**Ceritinib**	ALK—receptor TKI	ALK	NSCLC—ALK-positive in the first-line treatment or after prior crizotinib therapy	Bradycardia (2.3%) ^#^ QTc prolongation (1–3%) Cardiomyopathy (rare) ^#^	I
**Alectinib**	ALK—eceptor TKI	ALK, RET	NSCLC—ALK-positive in the first-line treatment or after prior crizotinib therapy	Bradycardia (8%) Cardiomyopathy (rare) ^#^	n/d
**Brigatinib**	ALK—eceptor TKI	ALK, ROS-1, IGF-1R	NSCLC—ALK-positive in the first-line treatment or after prior crizotinib therapy	Hypertension (20%) Bradycardia (8%) QTc prolongation (>1%) ^#^ Cardiomyopathy (rare) ^#^	n/d
**Lorlatinib**	ALK—receptor TKI	ALK, ROS-1	NSCLC—ALK-positive in the second-line treatment after prior ceritinib/alectinib or in subsequent lines after prior crizoninib and at least one other ALK inhibitor	Hypertension (13%) ^#^ Bradycardia (8%) QTc prolongation (0.7%) ^#^ Cardiomyopathy (rare) ^#^	n/d
**RET inhibitors**					
**Selpercatinib**	RET—receptor TKI	RET, VEGFR-1, -2, -3, FGFR-1, -2, -3	NSCLC—RET-positive second-line treatment after prior immunotherapy and/or platinum-based chemotherapy Thyroid cancer—RET-positive second-line treatment after prior sorafenib and/or lenvatinib	Hypertension (14–21%) QTc prolongation (6–15%)	n/d
**Epidermal growth factor receptor inhibitors**					
**Osimertinib**	Receptor TKI	EGFR, TKI-resistant mutation T790M	NSCLC—with activating EGFR mutations: - adjuvant treatment in stage IB-IIIA after complete resection - first-line treatment in advanced/metastatic setting - EGFR T790M mutation-positive advanced/metastatic setting	QTc prolongation (1.2–11%) Cardiomyopathy (3–5%)	n/d
**Lapatinib**	Receptor TKI	EGFR, HER2	Metastatic breast cancer HER2-positive - after prior therapy with anthracyclines, taxanes, and trastuzumab in metastatic setting: - in HR-negative patients in combination with trastuzumab after prior trastuzumab with chemotherapy - – in HR-positive patients in combination with aromatase inhibitor without indications for chemotherapy	QTc prolongation (rare) ^#^ Cardiomyopathy (2.2%)	I
**Mobocertinib ***	Kinase inhibitor of EGFR	EGFR exon 20 insertion mutation	NSCLC with EGRF exon 20 insertion mutation after prior platinum-based chemotherapy	QTc prolongation (1.2–11%) Cardiomyopathy (2.7%)	n/d
**BRAF/MEK inhibitors**					
**Vemurafenib**	Inhibitor of RAF kinases	RAF	Metastatic melanoma BRAF V600 positive	QTc prolongation (1–10%) ^#^ Cardiomyopathy	I
**Encorafenib**	Inhibitor of RAF kinases	RAF	Metastatic melanoma BRAF V600 positive in combination with binimetinib mCRC BRAF V600 positive—as a second-line treatment in combination with cetuximab	QTc prolongation (0.7–2.5%) ^#^ Cardiomyopathy (1–10%) ^#^	n/d
**Dabrafenib**	Inhibitor of RAF kinases	RAF	Metastatic melanoma BRAF V600 positive as monotherapy or in combination with trametinib Adjuvant treatment of melanoma BRAF V600 positive in stage III after complete resection in combination with trametinib NSCLC BRAF V600 positive in combination with trametinib	QTc prolongation (3%) ^#^ Cardiomyopathy (6%) ^#^	II
**Trametinib**	MEK inhibitor	MEK 1/2	Metastatic melanoma BRAF V600 positive as monotherapy or in combination with trametinib Adjuvant treatment of melanoma BRAF V600 positive in stage III after complete resection in combination with trametinib NSCLC BRAF V600 positive in combination with dabrafenib	Cardiomyopathy (11%)	III
**Cyclin-dependent kinase 4/6 inhibitors**					
**Ribociclib**	CDK 4/6 inhibitor	CDK 4/6	Metastatic breast cancer HR-positive, HER2-negative in combination with an aromatase inhibitor, or fulvestrant as a first-line treatment or a second-line treatment after prior endocrine therapy	QTc prolongation (7–16%)	n/d
**Abemaciclib**	CDK 4/6 inhibitor	CDK 4/6	Metastatic breast cancer HR-positive, HER2-negative in combination with an aromatase inhibitor, or fulvestrant as a first-line treatment or a second-line treatment after prior endocrine therapy	QTc prolongation (very rare) ^#^	n/d
**Palbociclib**	CDK 4/6 inhibitor	CDK 4/6	Metastatic breast cancer HR-positive, HER2-negative in combination with an aromatase inhibitor, or fulvestrant as a first-line treatment or a second-line treatment after prior endocrine therapy	QTc prolongation (very rare) ^#^	I
**BCR-ABL tyrosine kinase inhibitors**					
**Bosutinib**	Receptor TKI	BCR-ABL, Src, Lyn and Hck, PDGF, c-KIT	CML Ph-positive newly diagnosed in a chronic phase or after prior TKI in a chronic/blast phase	QTc prolongation (0.5%) ^#^ Fluid retention (1%)	I/II
**Dasatinib**	Receptor TKI	BCR-ABL, c-KIT, PDGFR-ß	CML Ph-positive newly diagnosed in a chronic phase or after prior imatinib in a chronic/accelerated/blast phase ALL and lymphoid blast CML in a second-line treatment	QTc prolongation (1%) ^#^ Cardiomyopathy (1.6%) Fluid retention (1%)	I
**Nilotinib**	Receptor TKI	BCR-ABL	CML Ph-positive newly diagnosed in a chronic phase or after prior imatinib in a chronic/accelerated phase	QTc prolongation (<5%) ^#^ Cardiomyopathy (<5%) ^#^	II
**Imatinib**	Receptor TKI	BCR-ABL, c-KIT, CSF-1R, PDGFRα, -β, DDR-1, -2	CML Ph-positive: - chronic phase for patients without indications for bone marrow transplant - chronic phase after failure of interferon-alfa therapy, or in an accelerated phase or blast crisis ALL Ph-positive first-line treatment in combination with chemotherapy, relapsed/refractory as monotherapy Myelodysplastic/myeloproliferative disease with PRGFR rearrangements Advanced hypereosinophilic syndrome and/or chronic eosinophilic leukemia with FIP1L1-PDGFR rearrangement GIST KIT (CD117)-positive: - metastatic setting - adjuvant setting at significant risk of relapse Dermatofibrosarcoma protuberans—unresectable/metastatic	Cardiomyopathy (1–2%) Fluid retention (1%)	II
**KIT/PDGFR inhibitors**					
**Ripretinib**	Receptor TKI	KIT, PDGFRα, -ß, TIE-2, VEGFR2, BRAF	GIST after prior three-line treatment including imatinib	Cardiomyopathy (1.2%–2.6%)	n/d
**FMS-like tyrosine kinase-3 inhibitors**					
**Gilteritinib**	Protein kinase inhibitor	FLT3, AXL	AML with FLT3 mutation- relapsed/refractory	Fluid retention (24%)	n/d

n/d, no data; * EMA approval is anticipated; compiled on the basis of the Food and Drug Administration label; For references, see the main text; ^ Adapted from [13]; ^#^ Complied on the basis of the Food and Drug Administration label; ^&^ For references, see the main text unless otherwise specified.

**Table 2 ijms-23-02815-t002:** Cardiotoxicity caused by selected kinase inhibitors together with suggested underlying mechanisms [29,30,34,35,36].

**Kinase Inhibitor**	**Cardiotoxicity**	**Suggested Mechanism**	**Reference**
TKIs used in kidney cancer	Hypertension	VEGFR inhibition NO and PGI_2_ synthesis inhibition Increase in ET-1 concentration Vasoconstriction Capillary rarefaction Reduction in vessel density	[34]
Ibrutinib	Ischemic heart disease Hypertension Bradyarrhythmias/atrial fibrillation	VEGFR2 inhibition NO formation inhibition Endothelial disfunction Vascular remodeling Inhibition of Src kinase Downregulation of PI3K-Akt pathway Cardiac fibrosis	[29,37]
Copanlisib	Hypertension	Mechanism remains unclear Vasoconstriction due to inhibition of PI3K (in the endothelium) suspected	[38,39]
Crizotinib	Sinus bradycardia	I_f_ inhibition Reduced current density of HCN4	[36]
Ponatinib	Myocardial infarction	AKT signaling pathway inhibition ERK inhibition Apoptosis induction	[30]
Dabrafenib	Cardiomyopathy	ERK inhibition Lower protection against oxidative stress	[40,41]
Trametinib	Cardiomyopathy	ERK inhibition Lower protection against oxidative stress CD47 transcription stimulation (inhibition of NO production) “Two-hit” hypothesis	[40,42]
Ribocyclib	qTc prolongation	Mechanism remains unclear Probably not due to CDK 4/6 inhibition The role of metabolites suspected	[43]
Imatinib	Fluid retention Cardiomyopathy	Mitochondrial dysfunction: reduction in mitochondrial membrane potential, cytochrome c release into the cytosol Activation of the ER stress response Cellular ATP content reduction	[35]

**Table 3 ijms-23-02815-t003:** Management strategy for TKI-induced hypertension according to the CTCAE grading system [7,58].

Arterial Hypertension (Grade)	Action
Grade 1 140–149/90–99 mmHg	Antihypertensive treatment Continuation of TKI therapy
Grade 2 160–179/100–109 mmHg	Antihypertensive treatment modifications Continuation of TKI therapy
Grade 3 ≥180/≥110 mmHg	Aggressive antihypertensive treatment Discontinuation of TKI therapy

## Data Availability

Not applicable.

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
