# Peer review of "Cardiotoxicity Induced by Protein Kinase Inhibitors in Patients with Cancer"

_ijms, 2022, doi:10.3390/ijms23052815_

Round 1

Reviewer 1 Report

The paper is well written and refers to a very common clinical situation – protein kinase inhibitors (KIs) used in cancer treatment and their cardiotoxic adverse effects.

Some minor adjustments are recommended, mostly in the sense of an easier reading or a more synthetic presentation of the findings. Some small inconsistencies were noticed.

R66 – KIs are classified into 20 families – but fig 1 includes 11 families and table 1 12 classes. Rewording may be needed. Also, keeping the same order of the classes in the test, in table 1, eventually in fig 1 could facilitate an easier understanding of the text.

Table 1 could be more illustrative if presented as landscape, with an additional column mentioning the class of KIs. The classes could be numbered, also for an easier reading.

Section 6 – molecular mechanisms – to be synthesized and eventually moved before section 4.

Author Response

Dear Reviewer,

Thank you for your useful comments. We addressed all of them in the text.

Please be informed that:

  • the text was corrected by a native speaker and corrected by the professional Medical English editor
  • findings were presented in a more synthetic way (eg. clinical tips were moved at the end of each cardiac adverse event section; Table 1 was updated and discussed in terms of cardiac adverse events incidence and significance)
  • Table 1 was also updated in terms of KIs classes as suggested.
  • Figure 1 was updated and is now consistent with other text as suggested, Figure 2 was added.
  • molecular section was extended, synthesized and moved before the clinical part to reflect the scope of the journal as suggested.

Yours sincerely,

Mirosława Puskulluoglu on behalf of the authors.

Reviewer 2 Report

Interesting paper, minor language modifications are needed

Author Response

Dear Reviewer,

Thank you for your kind comment.

Please be informed that in order to address yours ond other reviewers comments:

  • the text was corrected by a native speaker and by the professional medical English editor
  • findings were presented in a more synthetic way (eg. clinical tips were moved at the end of each cardiac adverse event section; Table 1 was updated and discussed in terms of cardiac adverse events incidence and significance)
  • Table 1 was also updated in terms of KIs classes as suggested.
  • Figure 1 was updated and is now consistent with other text as suggested.
  • Figure 2 was added.
  • molecular section was extended, synthesized and moved before the clinical part to reflect the scope of the journal as suggested.

Yours sincerely,

Mirosława Puskulluoglu on behalf of the authors.

Reviewer 3 Report

In the current review, the authors summarize the different types of cardiac adverse effects (AEs) induced by protein kinase inhibitors (KIs) in cancer patients. The main efforts were focused on the incidence rate of each AE recorded in tumor patients treated with KIs. Indeed, using the clinical trials data, as well as those in the observational, meta-analysis and retrospective studies, the authors assessed the frequency of KI-related cardiac complications, like hypertension, arrhythmias, fluid retention, ischemia and thromboembolic events, expressly in cancer.

Despite the review topic is in-trend, several concerns come out after a careful reading of the article. Firstly, the aim and scope of the IJMS journal are to provide an advanced forum for molecular studies in biology and chemistry, with a strong emphasis on molecular biology and molecular medicine. From this perspective, the current review is mainly focused on the KI-related clinical appearances, and thus it may not perfectly fit with the journal. Molecular and/or biological features of KIs are just marginally reported in the section 6. More emphasis should be given to this specific part in my opinion.

Concerning the contents, instead, some aspects are exhaustively addressed, while others should be further discussed. In this respect, table 1 surely represents a particular strength, providing a rapid and easy support to obtain information about the molecular targets and the clinical indications of each KI compounds employed in clinical. But, at the same time, the review lacks in critical comments and no innovative points of view are provided. These aspects are only marginally addressed in the discussion section indeed. In my view, even though different KIs are able to induce the same AE in the heart, the incidence rate is dissimilar and sometimes considerable. Discussing about this, as well as on the fact that that no clinical study has been designed to assesses the KI-induced cardiac dysfunction as a primary/secondary end-point, may further increase the review impact.  

Structurally, given that at the end of each subparagraph of the “cardiotoxicity of protein kinase inhibitors” section the authors include the strategy for managing the related AEs, table 2 should be moved immediately after 4.1. Hypertension. “Tips for clinicians” section is almost useless. It can be removed in its present form, otherwise it must be improved. Additionally, some references result to be quite old, especially those related to the chemotherapy-induced cardiotoxicity.

Minor Concerns:

Line 36: there is a redundancy in words “and the organization of healthcare care”

Line 63: a dash is wrongly typed in this line.

Line 115: “Its consequences have not yet been discovered [9,10]” it remains unclear the meaning of this sentence.

Line 135: References are missed.

Lines 140-146 seem out of place and quite unrelated with the previous argumentations. Please, include this part in a more fluent and blended way.

Line 156: there is a doble space in this line.

Line 223: Kis should be KIs

Line 234: there is a doble space in this line.

Line 375: this sentence seems unrelated to the previous ones.

Figure 1: design and conceptualization can be improved. In my view, including more colors or eliminating non-involved pathways’ components may stand out the KIs role at cellular stage. 

Since the authors examined the KI-induced AEs separately, it would be useful to provide an additional table where for each medication a list of the related cardiac AEs is reported.  

Author Response

Dear Reviewer,

Thank you for your useful comments and in-depth review.

We wanted to emphasize that we took your review really seriously and decided to re-build the whole manuscript to address all of your comments. In particular, to make the manuscript fitting more into IJMS molecular scope and to introduce critical comments..

The current sections of the manuscript are:

  • short introduction;
  • extended section presenting protein kinases (Fig 1 and 2) together with complementary Table 1 summarizing inhibitors of protein kinases (with suggested additional columns for cardiac AEs and classes of KIs);
  • shortened section presenting cardiotoxicity of systemic treatment as a complex topic and why it is different for traditional agents and targeted therapies
  • molecular mechanisms of KIs cardiac toxicity: starting from Table 2 (with exemplary drugs and possible involved pathways) that was extended and finishing with summaries of cardiac AEs one by one (hypertension, arrhythmias etc.) with discussion about molecular and cellular causes responsible for these AEs.
  • clinical data about cardiotoxicity of KIs; each section (hypertension, arrhythmias etc.) is now followed by short clinical suggestions.
  • conclusion section that was extended to emphasize:1. the findings from table 1 about different incidence of cardiac toxicity between different KIs 2. the difficulty with obtaining reliable clinical data from clinical trials due to cardiac safety not being primary endpoint in large studies 3. the challenges in presenting molecular and cellular mechanisms responsible for cardiac toxicity of KIs due to novelty of these compounds.

Please be also informed that:

  • The text was corrected by a native speaker and corrected by the professional medical English editor
  • Findings were presented in a more synthetic way (eg. clinical tips from previous section 5 were moved at the end of each cardiac adverse event section and the previous section no 5 was deleted; Table 1 was updated and discussed in terms of cardiac adverse events incidence and significance)
  • Table 1 was also updated in terms of KIs classes, which was also discussed in the text
  • Figure 1 was updated (split into Figure 1 and 2) and is now consistent with other text as suggested.
  • Molecular section was re-built and extended, synthesized and moved before the clinical part to reflect the scope of the journal as suggested.
  • All your minor comments were addressed
  • References were updated (e.g. by adding more recent ones regarding chemotherapy and removing outdated ones; by providing missing references; by updating new reference order)
  • The text was additionally reviewed by consultant in cardiology (information added in acknowledgment section)

Track changes was partially turned off while updating the table 1 to make it possible to follow the significant changes.

The main focus of this review is currently moved from clinical to molecular level. We truly hope that such in-depth analysis and reconstruction of the text is satisfactory and would result in your acceptance of the manuscript.

Yours sincerely,

Mirosława Puskulluoglu on behalf of the authors.

Round 2

Reviewer 3 Report

The revised form of the manuscript is surely improved in either content and structure. Nevertheless, minor revisions are required before publication:

A short overview about the role of KIs in cancer treatment is required in the introduction section.

Figure 1 should be further improved, following the suggested comments indicated in the previous report for instance. This schematic representation remains quite crowded, instead of Figure 2 that is clearly organized. 

A separate list of abbreviations, instead of an extended figures legend, should be useful to make the reading more fluent.

“5. Cardiotoxicity of protein kinase inhibitors” is quite disconnected from the subsequent subparagraphs. Please introduce the content of the following sections. 

Author Response

Dear Reviewer,

Thank you for your prompt reply and your comments regarding the revised version of the manuscript.

Please be informed that:

  • The sentence regarding the overview of the role of KIs in cancer treatment was added to the Intoduction section. It is later explored in Table 1 and column Therapeutic indications thus the Table 1 was again moved before the Figures1&2 (as initially). Also reference order for ref: 5-10 was changes and updated through the manuscript.
  • Figure 1 was improved according to your suggestions:it was simplified, colors were changed; abbreviations were updated;
  • Separate list of abbreviations was included (and Figure/Table legends were simplified/shortened). It was placed at the end as an additional Table (as we found that way of abbreviation placing in other IJMS published manuscripts)
  • A paragraph 5. was updated according to your suggestions starting now with a direct presentation of its subparagraphs.

New changes are marked now in green.

Yours sincerely,

Miroslawa Puskulluoglu on behalf of the authors